# Understanding the Influence of Personality Traits on Risk of Suicidal Behaviour in Schizophrenia Spectrum Disorders: A Systematic Review

**DOI:** 10.3390/jcm10194604

**Published:** 2021-10-08

**Authors:** Manuel Canal-Rivero, Rosa Ayesa-Arriola, Esther Setién-Suero, Benedicto Crespo-Facorro, Celso Arango, Rina Dutta, Javier-David Lopez-Morinigo

**Affiliations:** 1Hospital Universitario Virgen del Rocío, 41013 Sevilla, Spain; mcanalrivero@gmail.com (M.C.-R.); bcrespo@us.es (B.C.-F.); 2Centro de Investigación Biomédica en Red Salud Mental (CIBERSAM), 28029 Madrid, Spain; rayesa@idival.org (R.A.-A.); carango@hggm.es (C.A.); 3Grupo de Psiquiatría Traslacional, Instituto de Biomedicina de Sevilla (IBIS), 41013 Sevilla, Spain; 4Department of Psychiatry, Marqués de Valdecilla University Hospital, IDIVAL, School of Medicine, University of Cantabria, 39008 Santander, Spain; 5Department of Methods and Experimental Psychology, Faculty of Psychology and Education, University of Deusto, 48007 Bilbao, Spain; setiensuero@hotmail.com; 6Department of Child and Adolescent Psychiatry, Institute of Psychiatry and Mental Health, Hospital General Universitario Gregorio Marañón, IiSGM, CIBERSAM, School of Medicine, Universidad Complutense, 28029 Madrid, Spain; 7Institute of Psychiatry, Psychology and Neuroscience, King’s College London, London WC2R 2LS, UK; rina.dutta@kcl.ac.uk; 8South London and Maudsley NHS Foundation Trust, London SE5 8AZ, UK; 9Departament of Psychiatry, School of Medicine, Universidad Autónoma de Madrid, 28029 Madrid, Spain; 10Department of Psychiatry, Instituto de Investigaciones Sanitarias-Fundación Jiménez Díaz, 28029 Madrid, Spain

**Keywords:** personality traits, suicidal behaviour (SB), schizophrenia spectrum disorders (SSD)

## Abstract

Risk of suicidal behaviour (SB) in schizophrenia spectrum disorders (SSD) is a major concern, particularly in early stages of the illness, when suicide accounts for a high number of premature deaths. Although some risk factors for SB in SSD are well understood, the extent to which personality traits may affect this risk remains unclear, which may have implications for prevention. We conducted a systematic review of previous studies indexed in MEDLINE, PsycINFO and Embase examining the relationship between personality traits and SB in samples of patients with SSD. Seven studies fulfilled predetermined selection criteria. Harm avoidance, passive-dependent, schizoid and schizotypal personality traits increased the risk of SB, while self-directedness, cooperativeness, excluding persistence and self-transcendence acted as protective factors. Although only seven studies were retrieved from three major databases after applying predetermined selection criteria, we found some evidence to support that personality issues may contribute to SB in patients with SSD. Personality traits may therefore become part of routine suicide risk assessment and interventions targeting these personality-related factors may contribute to prevention of SB in SSD.

## 1. Introduction

Suicide is a major public health issue, which accounts for almost one million deaths every year across the world [1]. It is of major concern that suicide rates worldwide are very likely to significantly increase in the context of the COVID-19 pandemic [2,3,4]. Two well-known suicide risk factors will contribute to this. First, a rise in unemployment as a result of an economic turndown [5], which in previous economic recessions resulted in higher suicide rates [6]. Second, COVID-19 prevention measures are based on so-called physical distancing [7], which, although not inevitably [2], is likely to increase social isolation levels, a major suicide risk factor [8]. Suicide prevention should therefore become a public health priority in the post-COVID-19 pandemic years, which will require funding, significant effort, a multidisciplinary approach and international collaboration [9,10,11].

Every year almost two million people across the globe receive a first diagnosis of schizophrenia [12]. Schizophrenia is not only associated with poor clinical and social outcomes [13], but also with increased mortality [14]. Furthermore, the mortality gap between schizophrenia patients and the general population appears to have widened over the past few decades [15,16], and suicide has been demonstrated to be the strongest contributor to this excess mortality of schizophrenia [15,17]. Indeed, up to 5% of patients with schizophrenia die from suicide [18] and between 20% [19] and 40% of patients receiving mental healthcare who end their lives suffer from a psychotic disorder [20]. Risk of suicidal behaviour (SB) is particularly high in the first episode of psychosis (FEP) [15,21,22,23,24], although great caution is also needed ten years or longer after first presentation [23]. Specifically, there has been an increase in the incidence of reactive psychoses, i.e., a set of acute-onset and short-lived psychotic conditions triggered by psychosocial trauma [25]. In addition, social isolation-based measures for preventing COVID-19 were proposed to contribute to psychosis onset [26,27,28]. Moreover, so-called COVID-19-induced reactive psychosis was associated with an increased risk of suicidal behaviours [27].

In particular, two earlier meta-analyses examined risk and protective factors of SB in adults with schizophrenia [29] and in adolescents and adults with FEP [30]. Specifically, previous depression and suicide attempts, drug misuse, agitation or motor restlessness, fear of mental disintegration and poor medication compliance were found to increase suicide risk in schizophrenia [29]. In line with this, previous suicide attempts, expressed suicide ideation, greater insight, alcohol abuse, substance use, younger age of onset, younger age at first treatment, depressed mood and the duration of untreated psychosis were associated with an increased risk of deliberate self-harm before and after treatment onset in FEP [30]. On the other hand, hallucinations reduced suicide risk in schizophrenia [29]. In keeping with this, neither positive symptoms nor functioning were linked with an increased risk of deliberate self-harm in FEP [30].

This said, it should be noted that most of the studies included in the meta-analyses had a case–control design, in which odds ratios (ORs) measure the strength of the association between putative risk factors and a binary outcome, namely, SB (present vs. absent). Regardless of statistical significance, ORs < 2 were reported to indicate a ‘weak’ association. Interestingly, personality and psychosocial-related factors were found to be more strongly associated with SB, i.e., higher ORs, than biological variables [31].

Of relevance, unlike non-psychotic mental disorders (see other articles published in this special issue), the potential influence of personality traits on suicide risk in schizophrenia and in FEP were not included in the above meta-analyses [29,30], which was probably owing to the low number of available studies on this topic at that time. However, premorbid personality traits may play a role in the aetiology, course and outcome of psychotic disorders [32]. In keeping with this, recent FEP studies linked specific personality traits such as fearfulness, social inhibition, shyness, immaturity, impulsivity, aggression, vulnerability and lack of coping strategies with risk of SB [33].

We conducted this up-to-date systematic review with the aim of investigating the role of personality traits in SB, defined as “a self-inflicted, potentially injurious behaviour with a nonfatal outcome for which there is evidence (either explicit or implicit) of intent to die or self-inflected death with evidence (either explicit or implicit) of intent to die” [34], in schizophrenia spectrum disorders (SSD). In particular, two hypotheses were tested: (i) that the presence of personality traits such as social inhibition, shyness, impulsivity, aggression, vulnerability and lack of coping mechanism will be associated with an increased SB risk, and (ii) based on the aforementioned editorial [31], we postulated that statistics (such as ORs) measuring the strength of these associations will be greater than 2. Although not very common in systematic reviews, a hypothesis-driven approach has been strongly recommended in order to appraise the quality of systematic reviews’ methodology, that is, whether the research question was properly addressed [35].

## 2. Materials and Methods

We followed the Preferred Reporting Items for Systematic Reviews and Meta-Analysis (PRISMA) guidelines for the reporting of the findings of this systematic review [36].

### 2.1. Search Strategy

We conducted a literature search in MEDLINE, PsycINFO and Embase of articles that reported data on the relationship between personality traits and risk of suicidal behaviour (SB) in SSD patients, which encompasses suicide attempts and suicide completion. The terms included for the search were: “suic* OR self-harm” AND “personality traits OR temperament* OR character*” AND “psychos* OR psychot* OR schizophr*”. References from selected articles were cross-reviewed and selected if they met the following criteria.

### 2.2. Selection Criteria

All the abstracts from the initial search were independently screened by MCR and JDLM against the following selection criteria:(1)Peer-reviewed articles written in English from 1 January 1993 (when the ICD-10 [37] was published) to 15 February 2021, thus ensuring that the studies included would be based on similar definitions of SSD, which are detailed below;(2)Sample size of more than 10 patients (so case reports and case series were not included);(3)Age: 16–64 years, both inclusive, which was decided in order to include more FEP samples;(4)Diagnosis: ‘Schizophrenia spectrum disorders’, encompassing Schizophrenia, Schizoaffective Disorder and First-Episode Psychosis according to either ICD-10 [37] or DSM-IV-TR [38] or DSM-5 [39] definitions;(5)Clinical setting: In-, outpatients and mixed samples were included;(6)Study design: both cross-sectional and cohort studies were considered;(7)Outcome measures: data on suicide attempts and/or suicide completion had to be reported either before (cross-sectional studies) or after the study inception (cohort studies);(8)Either a premorbid or comorbid personality assessment had to be available;(9)No measure of the strength of the association between personality traits and SB had to be reported for the article to be included. We decided to include reports with no statistics since these studies may provide sufficient information to test hypothesis i.

In case of any doubt about meeting/not meeting the selection criteria, this would be independently resolved by two other authors (RAA and ESS), although no such discrepancies were identified between the two authors (MCR and JDLM) who conducted the articles search and selection; this was probably due to the low number of retrieved studies.

### 2.3. Data Extraction

Two authors (MCR and RAA) independently extracted all data by using a predetermined data extraction form and a web-based app for screening records in systematic reviews known as Rayyan [40]. Two other authors (ESS and JDLM) were independently contacted to resolve any inconsistencies as appropriate. The data extracted included first author and year of publication, country, setting (in- or outpatient), study design (cross-sectional or longitudinal/prospective/cohort), sample size and length of follow-up where applicable, diagnostic tool, diagnosis, personality traits assessment and the strength of the association between personality-related variables and suicidal behaviour outcomes, where available.

### 2.4. Primary Outcomes

The primary outcomes of this investigation were: (i) the association between personality traits and risk of SB (hypothesis i); and, if found, (ii) the strength of such an association (hypothesis ii).

Owing to the anticipated low number of published studies on this topic, including relevant between-study methodological differences (such as study design—cross-sectional vs. longitudinal studies—and outcomes measures), we could not apply meta-analytic techniques to results. Instead, we present a narrative review of the findings, although we used a semi-quantitative approach to test hypothesis ii (i.e., by quantitatively comparing the strength of the association between personality traits and SB across studies).

## 3. Results

### 3.1. Study Selection Process

The initial search yielded 6738 references, although after duplicate publications were removed n = 5114 articles were reviewed at a title/abstract level against the above selection criteria, following which n = 5091 were excluded. Twenty-three articles were therefore assessed for eligibility and so the full text was reviewed. Of these, seven papers fulfilled the selection criteria and were included in the systematic review. Of note, although qualitative studies were not excluded from the search, no such studies were identified. Figure 1, below, shows the PRISMA flow chart of the study selection process.

### 3.2. Characteristics of the Selected Studies

As detailed above, seven studies were included in the systematic review. Of these, four reported an association between specific personality traits and increased levels of SB. Of relevance, all the included studies analysed the relationship between personality traits and “suicide attempts”, i.e., no findings concerning the association of personality traits with suicide deaths were reported.

The first study analysed the relationship between personality traits and risk of suicide attempts in a sample of patients with SSD and was conducted in Turkey on a sample of n = 94 outpatients with schizophrenia (DSM-IV-TR criteria [38]) using a cross-sectional design [41]. The Temperament and Character Inventory (TCI) compared the Cloninger model factors between those with/without previous suicide attempts. After controlling for demographic and clinical variables, four personality features distinguished schizophrenia patients with/without history of SB. While harm avoidance (OR 7.59, 95%CI 0.65–0.93, *p* = 0.006) and persistence (OR 11.40, 95% CI 0.16–0.62, *p* = 0.001) scores increased the risk, self-directedness (OR = 6.77, 95% CI 1.05–1.38, *p* = 0.009) and cooperativeness (OR 5.88, 95%CI 1.04–1.46, *p* = 0.015) were protective factors [41]. These findings were partially replicated by a mixed (in- and outpatients) sample of n = 161 schizophrenia patients from Spain [42] in which, although only in males, the number of suicide attempts correlated negatively with self-directedness (OR 0.94, 95% CI 0.90–0.98, *p* < 0.05) and self-transcendence (OR 0.96, 95% CI 0.93–0.99, *p* < 0.05). The authors of the study hypothesised that these gender differences could be explained by the fact that males tend to struggle to regulate and adapt their behaviour, including self-reported lower levels of spirituality and universal identification than females.

A cross-sectional sample of n = 264 adults with schizophrenia from Croatia were evaluated using the Big-Five model of personality, although no personality dimension was linked with previous SB [43].

A case–control study was conducted in Japan in which n = 87 outpatients with schizophrenia were compared with n = 322 healthy controls. In particular, the personality traits of a subgroup of schizophrenia patients were evaluated using the Schizotypal Personality Questionnaire. Total scores as well as interpersonal and disorganized factors were higher (*p* < 0.01) in those with previous SB than in non-attempters [44].

Finally, a 1-year follow-up cohort of n = 65 FEP patients from Spain examined the relationship between personality traits and diverse characteristics of SB such as occurrence, temporality and frequency, that is, three studies aiming to answer different research questions, hence providing complementary information. First, SB over the six months after the psychosis onset was predicted by passive-dependent personality traits (OR 2.42, 95% CI 1.15–5.09, *p* = 0.02) [33], while the presence of SB at 1-year follow-up was related to schizoid personality characteristics (OR 1.62, 95% CI 1.02–2.57, *p* = 0.04) [45]. Personality traits, however, did not increase risk of repeated SB over the follow-up [46].

The characteristics of the selected studies are detailed in Table 1, below.

## 4. Discussion

### 4.1. Main Findings

We aimed to investigate the role of personality traits in risk of SB in SSD patients. First of all, only seven studies could be included in the systematic review. Although this may limit the generalisability of results, at the same time the low number of included studies suggests that this topic may have been somewhat neglected by research to date. Specifically, we predicted that personality issues would be linked with an increased risk of suicidal behaviour (SB) in SSD patients (hypothesis i) and we postulated that these associations would be relatively strong (i.e., ORs > 2). We found some evidence to support both hypotheses. In particular, harm avoidance, passive-dependent, schizoid and schizotypal personality traits emerged as the specific personality features associated with an increased risk of SB in schizophrenia, consistent with hypothesis i, while cooperativeness, excluding persistence self-directedness and self-transcendence behaved as protective factors, which was unexpected. It should be noted, however, that these findings only apply to suicide attempts, while the association of personality traits with suicide deaths remains to be investigated. Of relevance, despite the low number of selected studies, which may result in erroneously accepting the null hypothesis [33], we found some evidence of the association between personality traits and SB in SSD, which therefore provides further support for this finding. Moreover, in line with hypothesis ii, effect sizes were relatively large (most ORs > 2), as discussed further below.

### 4.2. Personality Issues Increased Risk of SB in SSD

Surprisingly, the extent to which personality traits may contribute to outcomes in patients with schizophrenia, including symptomatology, functioning or SB, has scarcely been analysed [47]. However, over the past few years there has been growing research interest in better understanding the temperamental and character features of SSD subjects, given their potential implications on outcomes [48], including SB [30].

Back in 1913, Jaspers [49] suggested “psychosis onset to be a process, that is, a permanent change in a person’s psychic life”; and this view appears to have dominated theoretical debate ever since, which may explain, in part, the very low number of identified studies in this systematic review. On the other hand, from a dimensional approach to psychosis, Kretschmer postulated that psychosis emerges from individual’s premorbid personality [50], which would therefore have an important bearing on outcomes [48]. Our results, which revealed that harm avoidance, self-directedness, passive-dependent as well as schizoid and schizotypal personality traits increase suicide risk in patients with schizophrenia would provide partial support for Kretschmer’s theories of “psychosis and premorbid personality as a continuum”. In other words, our findings appear to suggest that despite previous assertions to the contrary based on Jasper’s views, the onset of psychosis does not completely change premorbid personality traits, which in contrast, influence psychosis-related outcomes, including risk of SB.

More recently, the influence of personality traits on SB risk in patients with schizophrenia was investigated from three different, albeit overlapping, models [51]: (i) a five-factor model [52], (ii) the so-called psychobiological model of temperament and character [53], and (iii) the clinical or pathological personality model [54]. First, higher harm avoidance and persistence as well as lower self-directedness and cooperativeness were related to SB in a sample of schizophrenia patients [41]. Second, from a psychobiological model of temperament and character, partially replicating previous studies [41], self-directedness and self-transcendence was found to negatively correlate with SB [42]. To the best of our knowledge, only one previous study [43] employed an instrument, namely, the Big Five Inventory-10 based on five-model factor, to explore the potential relationship between SB and personality in a sample of patients with schizophrenia. Although no association between personality traits and SB emerged from the analyses, the study authors highlighted the potential role of neuroticism in suicide-related behaviours [43]. Finally, schizotypal [44], passive-dependent [33] and schizoid traits [45] increased SB risk in patients with schizophrenia [44] and FEP [33,45].

This being said, as expected (hypothesis ii), most of the ORs concerning the association between these personality traits and risk of SB took values well above 2. This finding should warn us further about the extent to which this topic has been neglected in research on suicide in psychosis. In particular, future studies should switch the focus from only researching putative biomarkers of SB to validation of more comprehensive suicide models, which, through the combination of biomarkers with psychosocial variables including personality traits, may perform better in terms of prediction. Our second hypothesis was based on a 2015 editorial, which provided a review of three decades of research on suicidal behaviour [31]. The authors highlighted the importance of the meaning of the strength of the association between two variables above and beyond conventional statistical “significance” at *p* < 0.05. More specifically, the authors of this editorial suggested that the association of personality traits with SB may reach higher ORs than those emerging from analysing the biological correlates of SB. On this basis, we postulated that the strength of the association between personality traits and risk of SB would be large, which was partially supported by our findings. However, it is worth noting that personality traits were measured with scores from routine scales and questionnaires, hence as continuous variables, in all the studies included in the systematic review. As a result, high/very high ORs concerning the relationship between personality traits measured as continuous variables (i.e., scores from scales and questionnaires) and SB may actually indicate a relatively weak association between the two.

Of relevance, systematic reviews including a low number of studies may increase the risk of erroneously accepting the null hypothesis [35], that is, the lack of association between personality traits and SB. Hence, the fact that we managed to find some evidence supporting the relationship between the two (hypothesis i) based on a low number of selected studies (7) strengthens further this novel contribution to the field.

Although up to approximately 50% of SSD patients were found to fulfil criteria for a personality disorder according to one previous study [55], as noted above, personality traits in schizophrenia have received little attention from researchers. In this regard, a previous prevalence study reported dependent, narcissistic, avoidant, schizotypal and schizoid personality traits to be the most common personality traits in FEP [56]. Interestingly, our findings showed that precisely these personality traits increased SB risk, hence these subjects may represent a high-risk group, particularly in early psychosis. However, to our knowledge there are no evidence-based treatments for personality issues in SSD, an area in which further research is warranted given their implications on outcomes.

Insight, which includes illness awareness, symptom relabelling and treatment compliance [57], was postulated to increase suicide risk in early psychosis [30], although three later independent FEP cohorts did not confirm this [22,58]. One may question whether specific personality traits may be linked with higher/lower levels of insight. In this regard, a 6-month follow-up FEP study linked more severe schizoid and sociopathic personality traits with poorer insight levels [48], which was replicated by a hierarchical reanalysis of predictors of insight with the same FEP cohort [59]. Interestingly, based on two independent studies, cooperativeness, which is linked with insight [60], in addition to excluding persistence [41] and self-transcendence [42], reduced suicide risk. Hence, cooperativeness, which is associated with insight [60], as a personality trait behaved as a protective factor for suicidal behaviour. This may lead to speculation that, in contrast to the commonly held view among clinicians, insight may prevent SB in psychosis [22,58]. Early intervention services should therefore prioritise treatments for improving some insight dimensions, namely, illness awareness and symptom relabelling [58], such as metacognitive training [61]. However, some difficulties should be acknowledged when restoring insight, which is likely to require gaining mastery over them [62].

Nevertheless, each of the above personality-related variables are based on a single study, with the exception of self-directedness, which was protective [41,42]. Hence, these findings should be taken with cautious while awaiting replication studies. On the other hand, the use of instruments based on different theoretical models did not prevent linking common personality features, such as undefined personal goals, external organization, low self-determination, autonomy and social withdrawal, with SB in patients with SSD [63]. In addition, impaired emotional communication led to hopelessness and social isolation in FEP patients, which may affect patient safety [64].

### 4.3. Strengths and Limitations

To our knowledge, this is the first systematic review on this topic, which is of major clinical relevance. In particular, we searched three major databases, although only seven articles fulfilled the above selection criteria. Hence, this piece of work makes a novel contribution to the field, which may have implications for suicide prevention, as discussed below.

However, this review has several limitations. First, the selection criteria may have been too restrictive and the search was limited to English. In addition, we only searched three major databases (i.e., MEDLINE, EMBASE and PsycInfo). Hence, this systematic review may not include all of the articles on this topic, particularly those published in the grey literature. Although only seven studies were selected and systematic reviews including a low number of records may increase the risk of erroneously accepting the null hypothesis [35], we managed to find some evidence supporting the link between personality traits and SB, that is, the alternative hypothesis. This noted, caution is needed when interpreting the findings in terms of their generalisability. For instance, some specific personality traits were only associated with SB in one study, which therefore require replication studies. Moreover, three out of seven selected studies were based on a relatively small FEP cohort (n = 65) [33,45,46]. However, as detailed above, results concerning personality issues and SB from these reports did not overlap each other. Second, we could not apply meta-analytic techniques to the studies’ results for the reasons explained previously. However, we took a semi-quantitative approach to the interpretation of the results, which allowed us to make a more reflexive presentation of the findings. Third, some sample sizes may have been too small, thus lacking sufficient statistical power to find an association between personality traits and SB and there were relevant differences in the personality and SB assessment. Fourth, the extent to which other variables acted as confounders/mediator was unclear from some analysed studies, which requires further investigation. Fifth, although only three articles using the same FEP cohort were identified [33,45,46], these reports actually tested different hypotheses. Finally, although not mandatory, PROSPERO registration of systematic reviews has been widely recommended over the past few years. Regretfully, we did not register the systematic review protocol at PROSPERO, although no differences in quality of research have been found between PROSPERO-registered and non-registered systematic reviews [65]. Nonetheless, this systematic review protocol is to be made retrospectively public in an open-access repository, such as Open Science Framework. In keeping with this, we did not evaluate the quality of selected studies, which aimed to answer different research questions. As a result, methods used by included studies varied very significantly, thus not allowing comparisons. This said, we detailed the characteristics and methodology of the seven included studies above.

### 4.4. Implications on Future Research and Clinical Practice

As noted above, only a few studies have investigated the contribution of personality issues to risk of SB in patients with SSD, and this risk is particularly high in early stages of the psychotic illness [21]. It is therefore envisioned that early intervention programmes may play a crucial role in preventing SB. Specifically, targeted interventions addressing specific risk factors and future studies measuring the extent to which these interventions reduce suicide rates in real-world conditions are warranted. Of note, caution is not only needed in early psychosis since risk of suicide in psychosis remains considerably higher a decade or even longer after first presentation [23].

In addition to well-known risk factors for suicide both in schizophrenia [29] and FEP [30], this systematic review revealed that, replicating findings from other mental disorders, personality appears to play a relevant role in suicide risk in SSD patients, which is the main contribution of this work. Therefore, not only personality traits should be considered as part of suicide risk assessment, but future trials testing interventions targeting harm avoidance, self-directedness, passive-dependent, schizoid and schizotypal personality traits, which may reduce suicide rates in psychosis, should also be carried out. This said, whether these personality traits may be amenable to modification is yet to be demonstrated.

In addition, the DSM-5, which was published in 2013 [39], and the upcoming ICD-11 [66] have incorporated dimensional models into the classification of psychotic and personality disorders [67,68]. Specifically, this approach should overcome the well-recognised limitations of Kraepelin’s first classification of psychotic disorders, on which previous DSM and ICD revisions were based [69]. From a research perspective, the DSM-5 and the ICD-11 may therefore pave the way towards linking specific symptoms and personality traits with biomarkers in SSD, that is, endophenotypes [70]. This may also lead to more targeted suicide risk assessment/management in SSD, a major challenge that remains unmet [19].

## 5. Conclusions

To sum up, this systematic review showed that some personality issues—harm avoidance, passive-dependent, schizoid and schizotypal personality traits—appear to increase risk of suicidal behaviour in patients with SSD, while other personality traits—cooperativeness, excluding persistence and self-transcendence—acted as protective factors. Our novel findings may have implications for suicide prevention in psychosis. The inclusion of personality traits may improve the predictive value of routine clinical suicide risk assessment in settings managing patients with psychosis, although this remains to be demonstrated. Also, interventions addressing harm avoidance, passive-dependent, schizoid and schizotypal personality traits may reduce suicide rates in psychosis, although future trials are needed to demonstrate this. Specifically, since risk is significantly higher in FEP, it is envisioned that early intervention services targeting suicide risk factors, including personality issues, may play a crucial role in suicide prevention in psychosis.

## Figures and Tables

**Figure 1 jcm-10-04604-f001:**
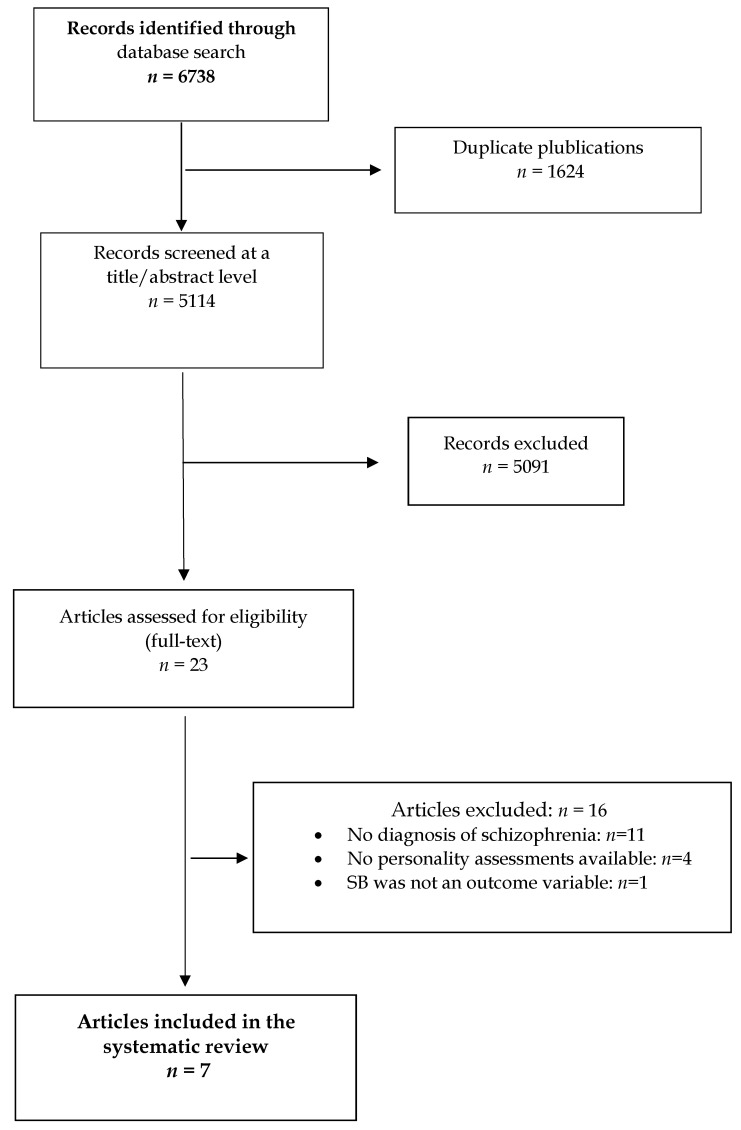
PRISMA flow chart of the study selection process.

**Table 1 jcm-10-04604-t001:** Characteristics of the selected studies.

Study	Country	Setting	Sample Size	Design	Follow-Up	Diagnosis	Personality Assessment	SB Assessment	Did Any Personality Trait Increase Risk of SB?	Statistical Results
Albayrak et al., 2012 [41]	Turkey	Out	94 Sch	C	N/A	Sch	TCI	Unstructured interviews/ clinical records	+	Cooperativeness: OR 5.88, 95%CI 1.04–1.46, *p* = 0.015; Harm Avoidance: OR 7.59, 95%CI 0.65–0.93, *p* = 0.006; Excluding Persistence: OR 11.40, 95% CI 0.16–0.62, *p* = 0.001; Self-Directedness: OR = 6.77, 95%CI 1.05–1.38, *p* = 0.009
Teraishi et al., 2014 [44]	Japan	Out	87 Sch322 HC	C	N/A	Sch	SPQ	Unstructured interviews/ clinical records	+	SA vs. nSA: Total SPQ score: 36.6 ± 12.5 vs. 26.6 ± 16.8, *p* = 0.005; SPQ Interpersonal factor: 17.4 ± 6.9 vs. 12.6 ± 8.1, *p* = 0.004; SPQ Disorganized factor: 8.7 ± 3.6 vs. 5.8 ± 4.4, *p* = 0.003
Miralles et al., 2014 [42]	Spain	In/Out	161 Sch214 HC	C	N/A	Sch	TCI-R	Unstructured interviews/ clinical records	-	Self-directedness (OR 0.94, 95% CI 0.90–0.98, *p* < 0.05) and Self-Transcendence (OR 0.96, 95% CI 0.93–0.99, *p* < 0.05) negatively correlated with number of previous suicide attempts.
Canal-Rivero et al., 2016 [33]	Spain	In	65 FEP	L	12 months	FEP	PAS	SCAN	+	Passive-dependent personality traits associated with first suicide attempt occurred 6-month after FEP OR 2.42, 95% CI 1.15–5.09, *p* = 0.02
Canal-Rivero et al., 2017 [45]	Spain	In	65 FEP	L	12 months	FEP	PAS	SCAN	+	Schizoid personality traits associated with suicide attempts after FEP OR 1.62, 95% CI 1.02–2.57, *p* = 0.04
Jovanovic et al., 2019 [43]	Croatia	In	264 Sch	C	N/A	Sch	BFI-10	SIBQ	-	N/A
Canal-Rivero et al., 2019 [46]	Spain	In	65 FEP	L	12 months	FEP	PAS	SCAN	-	N/A

SB: Suicidal Behaviour; In: Inpatient; Out: Outpatient; Sch: Schizophrenia; HC: Healthy Controls; FEP: First Episode Psychosis; C: Cross-Sectional; L: Longitudinal; TCI: Temperament and Character Inventory; SA: Suicide Attempts; SPQ: Schizotypal Personality Questionnaire; TCI-R: Temperament and Character Inventory-Revised; PAS: Personality Assessment Schedule; BFI-10: Big Five Inventory-10; SCAN: Schedules for Clinical Assessment in Neuropsychiatry; SIBQ: The suicide Ideation and Behaviour Questionnaire; +: There is a relationship; -: There is no relationship; OR: Odds Ratio; CI: Confidence interval; nSA: non-suicide attempt. N/A: Not applicable.

## Data Availability

Datasets are available upon request, provided data policy access is complied with.

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
