# Peer review of "Understanding the Influence of Personality Traits on Risk of Suicidal Behaviour in Schizophrenia Spectrum Disorders: A Systematic Review"

_jcm, 2021, doi:10.3390/jcm10194604_

Round 1

Reviewer 1 Report

The authors adequately addressed this reviewers' concerns.

Reviewer 2 Report

Thank you for the opportunity to review the manuscript. This is a very well written paper dealing with the influence of personality traits on suicidal behaviours in people with schizophrenia spectrum disorders. This is an important topic with clear implications for clinical practice and policy making of support provision for people experiencing psychosis and suicidality. The importance of the present review is increased by the scant research on the subject. The focus on both suicide attempts and completed suicide (i.e., suicide death) needs to be clearly highlighted throughout the paper.

Minor comments and questions are listed below.

Introduction:

  1. A definition of ‘reactive psychoses’ would be useful in the introdution.
  2. Regarding the sentence, ‘Most importantly, the COVID-19 pandemic and its prevention measures represent a risk factor for reactive psychoses, which are characterised by suicidal ideation/behaviour from first presentation [9].’, can you give an example of COVID-19 prevention measures as a risk factor for people with psychosis?
  3. Please give examples of schizophrenia related disorders (line 50).
  4. It would be helpful if you could give a definition of suicidal behaviour. Are you including suicide attempts or self-harm with or without intention to die, for example?
  5. What protective factors did the two meta-analyses (references 27 and 28) examine?
  6. In hypothesis 1, what do you consider a ‘patholgical personality trait’?
  7. I feel the manuscript is missing some information regarding the relationships between personality, psychosis and sucidal behaviours. In the abstract you stated that ‘the extent to which personality traits may affect this [SB] risk remains unclear…’. I assume there is some research on this topic and wondered whether you could include and discuss any of these studies in your introduction. Some more detail on the findings of studies 31 and 32 about the link between personality traits and suicide attempts would strengthen the raitonale.

Materials and Methods:

  1. You have included suicide completion in the definition of suicidal behaviours but this is not reflected in the study title, hypotheses, or in the introduction. Personality risk factors may be different for suicidal behaviours and suicide death. Please specify in the introduction that the focus is on the relationships between personality traits and suicide death, as well as suicidal behaviours.
  2. For ease of reading, I would write SA, SC, SB and SSD in full throughout the paper.
  3. It is not clear why you have chosen to include people aged between 16 and 64 years.
  4. Were qualitative studies included in the review? If not, please provide a rationale.
  5. Point 10) does not seem like a study selection criterion. How were study selection disagreements resolved between the authors? Please specify why you did not conduct an inter-reater relibility analysis between the two authors conducting study screening and selection.
  6. Please provide a rationale for not conducting a quality appraisal of the included studies.

Results:

  1. Please specify in your results whether the findings of each study in relation to suicidal behaviours refer to suicide attempts or suicide death, or both.
  2. If possible, please explain what the authors of study referenced 36 mean by ‘suicidal’ and ‘non-sucidal’.
  3. The findings of study referenced 40 relate to men. Were other genders excluded from the study? A comment on the reaons why these findings may be specific to men would be useful.
  4. The findings on lines 206 and 207 state that self-directedness was a protecitve factor as reported by Albayrak et al. but in the abstract (line 29) and discussion (lines 245-247), self-directedness is reported as a risk factor for suicidal behaviours.

Table 1:

  1. Please add the reference number next to the author name in the first column.

Discussion:

  1. Based on the findings, can you identify which personality traits increased/decreased the risk of suicide death and suicide attempts, separately?

Author Response

This manuscript is a resubmission of an earlier submission. The following is a list of the peer review reports and author responses from that submission.

Round 1

Reviewer 1 Report

The present work does not contain the information and detailed description of the materials and methods required for the publication in this journal. Moreover, in my opinion some procedures are not correct, In addiction, results provided are not discussed in detail. Thus, I do not recommend to publish it on Journal of Clinical Medicine.

Reviewer 2 Report

This is a well designed study into the connections between personality traits and a risk for a fatal outcome (suicide) in schizophrenia patients.

The rationale is well presented, methods appear sound and well-explained, and, perhaps, most importantly, the results are critically presented with appropriate review of strengths and weaknesses.

Reviewer 3 Report

Strengths:  The manuscript addresses an important but understudied area.  The weaknesses are clearly described in the strengths and limitations section.

Weaknesses:  While these are noted in the strengths and limitations section, some of these are major weaknesses.  The review only describes 7 studies and only 4/7 found relationships between personality traits and SB.  Additionally, 3/7 studies were conducted by the first author.  3/7 studies appear to be based on the same sample of 65 subjects.

The Introduction does not support Hypothesis 2 (weak association), although this was found later in the paper.  I would suggest mentioning this earlier.

Hypothesis 2 us based on the strength of association but no measure of strength of association was required to be included in the review.

With the scant number of studies included in the review, it is difficult to draw conclusions, especially when combining dimensions such as temperament and personality disorder diagnostic symptoms.